VirION2: a short- and long-read sequencing and informatics workflow to study the genomic diversity of viruses in nature

Zablocki Olivier 1 2
Michelsen Michelle 3
Burris Marie 1
Solonenko Natalie 1
Warwick-Dugdale Joanna 3 4
Ghosh Romik 1
Pett-Ridge Jennifer 5
Sullivan Matthew B. 1 2 6
Temperton Ben b.temperton@exeter.ac.uk 3
1 Department of Microbiology, The Ohio State University , Columbus , OH , United States of America
2 Center of Microbiome Science, The Ohio State University , Columbus , OH , United States of America
3 School of Biosciences, University of Exeter , Exeter , Devon , United Kingdom
4 Plymouth Marine Laboratory , Plymouth , Devon , United Kingdom
5 Physical and Life Sciences Directorate, Lawrence Livermore National Laboratory , Livermore , CA , United States of America
6 Department of Civil, Environmental and Geodetic Engineering, The Ohio State University , Columbus , OH , United States of America
Rappe Michael
Electronic publication date: 2021 Mar 30
Publication date: 2021
Volume: 9
Electronic Location ID: e11088
Received 2020 Nov 6; Accepted 2021 Feb 19
Copyright: ©2021 Zablocki et al.
Copyright year: 2021
Copyright holder: Zablocki et al.
License: This is an open access article distributed under the terms of the Creative Commons Attribution License, which permits unrestricted use, distribution, reproduction and adaptation in any medium and for any purpose provided that it is properly attributed. For attribution, the original author(s), title, publication source (PeerJ) and either DOI or URL of the article must be cited.
License URL: https://creativecommons.org/licenses/by/4.0/

Keywords: Viral metagenomics, Virus, Virome, Metagenome, Nanopore sequencing, Phage, Long-reads

Funding: The U.S. Department of Energy, Office of Biological and Environmental Research, Genomic Science Program ‘Microbes Persist’ Scientific Focus Area #SCW1632 The U.S. Department of Energy, Office of Science, Office of Biological and Environmental Research, Genomic Science Program DE-SC0020173 The Gordon and Betty Moore Foundation #5488 The Natural Environment Research Council GW4+ Doctoral Training Program NE/L002434/1 NERC NE/R010935/1 Simons Foundation BIOS-SCOPE program The Wellcome Trust Institutional Strategic Support Fund WT097835MF Wellcome Trust Multi User Equipment Award WT101650MA BBSRC LOLA award BB/K003240/1 Auspices of DOE Contract DE-AC52-07NA27344 Olivier Zablocki, Matthew B. Sullivan and Jennifer Pett-Ridge were supported by the U.S. Department of Energy, Office of Biological and Environmental Research, Genomic Science Program ‘Microbes Persist’ Scientific Focus Area (award #SCW1632). Matthew B. Sullivan is supported by the U.S. Department of Energy, Office of Science, Office of Biological and Environmental Research, Genomic Science Program under Award Number DE-SC0020173 and by the Gordon and Betty Moore Foundation (grant #5488). The efforts of Joanna Warwick-Dugdale were funded by the Natural Environment Research Council GW4+ Doctoral Training Program (NE/L002434/1). Michelle Michelsen and Ben Temperton were funded by NERC (NE/R010935/1), with additional funding for Ben Temperton provided by the Simons Foundation BIOS-SCOPE program. This project used equipment funded by the Wellcome Trust Institutional Strategic Support Fund (WT097835MF), Wellcome Trust Multi User Equipment Award (WT101650MA) and BBSRC LOLA award (BB/K003240/1). Work at Lawrence Livermore National Laboratory was conducted under the auspices of DOE Contract DE-AC52-07NA27344. The funders had no role in study design, data collection and analysis, decision to publish, or preparation of the manuscript.

==============================
Microbes play fundamental roles in shaping natural ecosystem properties and functions, but do so under constraints imposed by their viral predators. However, studying viruses in nature can be challenging due to low biomass and the lack of universal gene markers. Though metagenomic short-read sequencing has greatly improved our virus ecology toolkit—and revealed many critical ecosystem roles for viruses—microdiverse populations and fine-scale genomic traits are missed. Some of these microdiverse populations are abundant and the missed regions may be of interest for identifying selection pressures that underpin evolutionary constraints associated with hosts and environments. Though long-read sequencing promises complete virus genomes on single reads, it currently suffers from high DNA requirements and sequencing errors that limit accurate gene prediction. Here we introduce VirION2, an integrated short- and long-read metagenomic wet-lab and informatics pipeline that updates our previous method (VirION) to further enhance the utility of long-read viral metagenomics. Using a viral mock community, we first optimized laboratory protocols (polymerase choice, DNA shearing size, PCR cycling) to enable 76% longer reads (now median length of 6,965 bp) from 100-fold less input DNA (now 1 nanogram). Using a virome from a natural seawater sample, we compared viromes generated with VirION2 against other library preparation options (unamplified, original VirION, and short-read), and optimized downstream informatics for improved long-read error correction and assembly. VirION2 assemblies combined with short-read based data (‘enhanced’ viromes), provided significant improvements over VirION libraries in the recovery of longer and more complete viral genomes, and our optimized error-correction strategy using long- and short-read data achieved 99.97% accuracy. In the seawater virome, VirION2 assemblies captured 5,161 viral populations (including all of the virus populations observed in the other assemblies), 30% of which were uniquely assembled through inclusion of long-reads, and 22% of the top 10% most abundant virus populations derived from assembly of long-reads. Viral populations unique to VirION2 assemblies had significantly higher microdiversity means, which may explain why short-read virome approaches failed to capture them. These findings suggest the VirION2 sample prep and workflow can help researchers better investigate the virosphere, even from challenging low-biomass samples. Our new protocols are available to the research community on protocols.io as a ‘living document’ to facilitate dissemination of updates to keep pace with the rapid evolution of long-read sequencing technology.

Introduction

Microbes are recognized as a major driving force in the functioning and maintenance of most ecosystems (Cavicchioli et al., 2019); however, research in the past decade suggests viruses—mostly those that infect bacteria (‘phages’)—are equally important. In the world’s oceans, viruses (mostly dsDNA phages) modulate microbial gene flow and are integral to global oceanic nutrient cycles (Brum et al., 2015; Roux et al., 2016; Gregory et al., 2019). Because most viruses are uncultivated, advances have mainly arisen through metagenomic approaches, which have rapidly improved with new sequencing technologies (Brum & Sullivan, 2015). The importance of viruses in community composition and nutrient cycling is also increasingly being recognized across diverse ecosystems, including soils (Emerson et al., 2018; Trubl et al., 2018), the human microbiome (Shkoporov et al., 2019), glacial ice (Zhong et al., 2020), and invertebrates (Shi et al., 2016; Wolf et al., 2020). For example, viruses have the potential to aid in soil carbon flux by encoding plant polysaccharide-degrading enzymes (Emerson et al., 2018), or can be involved in human gut dysbiosis that leads to various health issues (Mirzaei & Maurice, 2017).

Just as the scalability afforded by Illumina over 454 sequencing catapulted viromics from “gene ecology” to “population-based or genome-resolved ecology” (Brum & Sullivan, 2015), long-read sequencing offers promise for another transformative step forward (Warwick-Dugdale et al., 2019). Recent evidence suggests that current short-read-sequencing metagenomic methods are not capturing the whole of the virosphere. Within dsDNA viruses, the most extensively studied, short-read metagenomes capture abundant viruses at the taxonomic level of species or genera, but not likely genotypes. This is because complex virus communities contain mixtures of strains (i.e., virus population variants at the nucleotide level), but assemblers cannot reconstruct individual strains—instead strain mixtures are collapsed into a single ‘consensus’ genome or multiple genome fragments (Nurk et al., 2017), thus masking strain-specific features that indicate the functional diversity within viral community members (Nelson et al., 2016). Though the extent of populations missed is unknown, two separate studies –one using single-cell genomics (Martinez-Hernandez et al., 2017) and the other long-read sequencing (Warwick-Dugdale et al., 2019)—have demonstrated that strain-level diversity is under-represented in short-read datasets (with 3-fold higher nucleotide diversity captured in long-read viral metagenomes compared to short-read viral metagenomes; (Warwick-Dugdale et al., 2019). Specifically, both studies demonstrate that high genome microdiversity (i.e., the level of intra-population nucleotide variants for a virus ‘species’) could act as a barrier to full genome assembly, while masking co-occurring population variants. Moreover, emerging research suggests that hybrid assemblies (i.e., short- and long-reads co-assembly) could potentially recover more genomic diversity in natural samples (Cook et al., 2020).

We previously introduced VirION (Warwick-Dugdale et al., 2019), a custom library sequencing and informatics workflow that leveraged the strengths of both short- and long-read sequencing (using Oxford Nanopore Technologies –‘ONT’). Briefly, this first entailed shearing of the sample DNA to ∼8kb fragments, followed by DNA amplification through ‘Linker Amplified Shotgun Library’ (LASL; Breitbart et al., 2002; Duhaime et al., 2012) with a high-fidelity polymerase and 15 PCR cycles to minimize rates of chimera formation. These steps produced sufficient DNA to meet the minimum DNA input requirement (1 µg) for ONT sequencing. Long-read sequencing data was then assembled with an overlap layout consensus strategy (via Canu; Koren et al., 2017), and contigs were error-corrected with short-reads (with Pilon; Walker et al., 2014). Applying the full VirION workflow enabled a two-fold increase in capturing ‘complete’ genomes and improve both the length and number of recovered niche-defining hypervariable islands. However, the input DNA requirement (∼100 ng extracted from 20 L of seawater) for VirION is a barrier to generating long-read virome data from samples that yield little viral DNA, either due to challenges in extraction and/or low-volume, high resolution sampling. Furthermore, since the introduction of VirION, additional long-read assemblers and error-correction tools have been introduced, which have not been evaluated for virus metagenomic datasets. Here, we introduce VirION2, which includes significant wet-lab and analytical optimizations at key protocol steps to reduce input DNA requirements and increase long-read lengths and accuracy. We applied this new workflow to a concentrated natural seawater virioplankton community to assess VirION2’s effectiveness at capturing community features relative to VirION and other short-read approaches.

Materials and Methods

Phage mock community, DNA extraction and short-read sequencing

Three Pseudoalteromonas phage isolates (PSA-HM1, PSA-HP1 and PSA-HS2; see Table S1 for more details) were used as a mock community. Each phage was grown in culture and genomic DNA extracted as previously described (Duhaime et al., 2017). To produce the mock community, equimolar DNA concentrations from each phage extract were mixed. This mixture was used as template for sequencing library preparation in all sequencing runs for optimizing VirION2 library preparation. Separately, short-read sequencing libraries of the same mock community was prepared with NEXTFLEX® Rapid DNA-Seq Kit 2.0 (Bio Scientific Corp, cat#NOVA-5188-01). Sequencing was performed at the University of Exeter on a HiSeq 2500 instrument running the 2 x125 bp paired-end configuration.

Library preparations prior to Nanopore sequencing

Three DNA library preparations were used: (1) Direct sequencing of DNA with no PCR amplification and no shearing of viral DNA (referred to as ‘unamplified’), followed by the manufacturer’s sequencing library protocol (using ligation sequencing kit SQK-LSK109; see below for sequencing library preparation details). (2) The VirION pipeline, performed as previously described (Warwick-Dugdale et al., 2019). Briefly, this entailed shearing of high-molecular weight DNA, followed by ligation of amplification primers and PCR amplification of the sheared DNA prior to applying the manufacturer’s library protocol for sequencing. (3) VirION2, performed as follows: From the DNA extract, 10 µl were set aside for short-read sequencing, and the remainder was sheared to 10 kbp or 15 kbp fragments with Covaris g-TUBE according to the manufacturer’s instruction (except for the 15 kbp treatment, in which the samples were spun at 2, 075 × g for 60 s). Fragmented DNA was repaired and dA-tailed using the NEBNext FFPE DNA Repair Mix and NEBNext Ultra II End Repair/dA-Tailing Module, according to the manufacturer’s instructions except incubation time, which was reduced from thirty to five minutes. The repaired DNA was cleaned using Ampure Beads (Beckman Coulter) according to the manufacturer’s instructions (except for the final incubation time, in which 55 °C was used instead of room temperature) at a 1:1 ratio (v/v) and eluted in nuclease-free water. Ligation of barcoded adapters (Oxford Nanopore PCR Barcoding Expansion 1-12 kit, cat# EXP-PBC001) used for linker-amplified shotgun library (LASL) amplification was implemented as follows: A reaction mix composed of 50 µl NEB Blunt / TA Ligase master mix, 20 µl of the Oxford Nanopore barcode, and 30 µl of cleaned DNA was incubated for ten minutes at room temperature. The DNA was then immediately cleaned with Ampure beads with 1:0.4 sample to bead ratio to remove short fragments, and eluted in 15 µl nuclease-free water at 55 °C. PCR amplification on the cleaned libraries was performed as follows: A reaction mix using LA TaKaRa Hot Start kit (Takara Bio USA) was prepared according to the manufacturer’s instructions, using 5 µl of cleaned DNA and 2 µl of the desired barcode (from the same Oxford Nanopore barcoding kit, as previously). Cycling conditions were 94 °C for 1 min, 15 cycles of 94 °C for 30 s, 62 °C for 30 s, 68 °C for 8-16 min, and final elongation for 8-16 min at 72 °C. Elongation times varied according to the desired amplicon size: 8 min for 10 kbp; 12 min for 15 kbp; and 16 min for 20 kbp. Amplicons were subsequently cleaned with Ampure beads using a 1:0.5 sample to bead ratio, and eluted in 20 µl nuclease-free water. The cleaned amplified libraries were then used as input for the 1D genomic DNA by ligation kit (SQK-LSK109, Oxford Nanopore Technologies) according to the manufacturer’s instructions, with some modifications. In the ‘DNA repair and end-prep’ step, DNA CS (a standard DNA sequence used as a positive/quality control by ONT) was excluded, and instead, 48 µl of input DNA was added. In addition, incubation temperature was increased to 25 °C from 20 °C. In the ‘Adapter ligation and clean-up’ step, the Long Fragment Buffer (‘LFB’) was used to enrich for longer fragments. In the bead resuspension step, pellet resuspension was incubated for 10 min at 55 °C (instead of room temperature) since better DNA dissociation from the beads was observed to be more efficient at a higher temperature (data not shown). The remainder of the protocol was unchanged.

Mock community sequencing tests

We tested four high-fidelity and/or long-range DNA polymerases in duplicate to evaluate improvements of read length during the amplification step: (1) NEBNext (NEB M0541), (2) NEB Q5 (NEB M0491), (3) NEB LongAmp (NEB M0287) and (4) TaKaRa LA Taq (TaKaRa RR042). For each library, 15 amplification cycles were used and input DNA (mock community phage genomes) was sheared at 10 kbp (using Covaris g-TUBE). Influence of initial shearing length was also tested at 15 kbp with a DNA input of 80 ng/µl. For TaKaRa LA Taq, we also tested a shearing length of 15 kbp using 1 ng of input DNA to evaluate efficiency of VirION2 when minimal input DNA was available. For PCR cycling tests, 1 ng of input DNA sheared at 15 kbp was used as template.

Preparation of viral DNA from the Western English Channel

20 L of seawater from the Western English Channel (‘WEC’) was collected in rosette-mounted Niskin bottles at a depth of 5m from the Western Channel Observatory (WCO; http://www.westernchannelobservatory.org.uk/) coastal station ‘L4’ (50°15.00N; 4°13.00W) on 11th February 2019. Seawater was placed in a coolbox at ambient temperature and upon return to shore, was transported to the University of Exeter for processing within six hours of collection. The cellular fraction was removed via sequential filtration through glass fiber (GF/D: pore size 2.7 µm) and polyethersulfone (pore size 0.22 µm) filters in a 142 mm polycarbonate rig with a peristaltic pump. Viruses were precipitated and concentrated from the filtrate by iron chloride flocculation and collected on 1.0 µm polycarbonate filters (John et al., 2011). The viruses were immediately resuspended in ascorbate-EDTA buffer (0.1 M EDTA, 0.2 M MgCl2, 0.2 M ascorbic acid, pH 6.0) using two mL of buffer per 1 L of seawater. The resuspended viral fraction was transferred equally to four Amicon Ultra 100 kDa centrifugal filter units (Millipore UFC910024) which had been pre-treated with 1% bovine serum albumin buffer to reduce capsid-filter binding (Deng et al., 2014) and flushed with SM buffer (0.1 M NaCl; 0.05 M Tris–HCl; 0.0008 M MgCl2). The resuspended viral fraction was concentrated to 500–600 µl and removed from the filter unit; the Amicon filters were then washed with 200 µl of SM buffer (Bonilla et al., 2016) to ensure resuspension of all viral particles from capsid-filter adhesion. The viral fraction was purified with DNase 1 (100 U/mL; 2 h at room temperature) to remove non-encapsulated DNA. DNase 1 activity was terminated by the addition of 0.1 M EGTA and 0.1 M EDTA (Hurwitz et al., 2013). Viral DNA was extracted from the concentrated and purified viral fraction using Wizard DNA Clean-Up System (Promega A7280) to remove PCR inhibitors (John et al., 2011). The viral DNA was cleaned and concentrated by a 1.5 Ampure bead cleanup for downstream application.

Long and short read processing of Western English Channel viral DNA

Viral DNA from the WEC was used to prepare three separate long-read sequencing libraries and one short-read only library for comparison and error correction. (1) VirION libraries were prepared as described previously (Warwick-Dugdale et al., 2019), using 100 ng of input DNA; (2) Unamplified sequencing of viral DNA was performed using ∼3 µg of unsheared input DNA in a LSK-SQK109 library preparation (ONT) for genome sequencing according to the manufacturer’s instructions, and sequenced on a MinION R9.4 revD flowcell for 48 h. (3) A VirION2 library was prepared by first treating the DNA with the Zymo DNA Clean & Concentrator Kit (cat# D4013) according to the manufacturer’s instructions. The clean concentrate (90 µl, final concentration of 14.6 ng) was then sheared into 15 kbp fragments, used as template for 15 PCR cycles, and processed for MinION sequencing on a single flowcell according to the manufacturer’s instructions with the modifications stated in the previous section (“Library preparations prior to Nanopore sequencing”); (4) Short Read sequencing was performed using 51 ng of input DNA in a 1S Plus (Swift Biosciences) library preparation and sequenced to a depth of ∼25M 2 × 125 bp paired end sequencing on a HiSeq 2500 at the University of Exeter.

Long-read processing and quality control

Raw fast5 files containing ONT long-reads were basecalled with Guppy v2.3.1 (provided by Oxford Nanopore), using the flip-flop model. Reads were allocated into barcode bins by porechop (Wick, 2017), using –require_two_barcodes, –discard_unassigned and –discard_middle parameters in order to limit library cross-talk in multiplexed samples, and to remove concatenated reads where two strands pass through the same pore in quick succession. NanoFilt 2.2.0 (De Coster et al., 2018) was used to trim the first 50 bases of reads (to ensure any residual barcode sequence removal) and remove reads <1 kbp or with a phred quality score <9. Chimeric PCR products were identified for quantification and subsequent removal from each demultiplexed run using yacrd (Marijon, Chikhi & Varré, 2020) with default parameters. Remaining reads were kept for further analysis. To compare read lengths between mock community datasets, we randomly subsampled the reads (with replacement) using the ‘sample’ package in R version 3.5.0. Differences in bootstrapped medians (n = 50,000, 1000 replicates) between the barcoded libraries associated with each treatment (DNA polymerase type, DNA shearing size and PCR cycling number) were plotted as boxplots with the R package ‘ggpubr’.

Assessing assembly and error profiles

For testing assembly performance of the WEC sample, reads that passed quality control (described above) were first sub-sampled using bbtools reformat (Bushnell, 2015) in order to mitigate library size bias in assembly comparisons. Sub-sampling was based on the number of reads in the smallest library (500,000 reads in the unamplified dataset). We compared median read length between the full datasets and the sub-sampled datasets to confirm subsampling had not biased read length distributions (Fig. S1), and observed negligible shifts between full and sub-sampled datasets in both VirION (median read length: 3,020–3,023 bp 95% CI and 3,018–3,027 bp 95% CI) and VirION2 datasets (3,906–3,916 bp 95% CI and 3,900–3,917 bp 95% CI). Subsampled reads were used in three assembly approaches to determine optimal assembly strategy: (1) Overlap-layout consensus assembly (‘OLC’): All-vs-all alignments were generated using minimap2 v2.17-r941 (Li, 2018), and used to build an assembly with Miniasm v0.3-r179 (Li, 2016). Minipolish v0.1.2 (Wick & Holt, 2020) was used to iteratively apply Racon-based contig polishing (Vaser et al., 2017) on Miniasm assemblies. Polished OLC assemblies in GFA format were converted to FASTA using the following awk command: “ ‘$1 ∼/S/ {print “>”$2“\n”$3}’ assembly.gfa >assembly.fasta”. Fasta-formatted Racon assemblies were further error-corrected with Medaka v0.11.5 (Oxford Nanopore Technologies Ltd, 2018), with an appropriate model select to correspond to the sequencing chemistry and basecaller model used. Lastly, Pilon v1.23 (Walker et al., 2014) was used to correct errors using short-read mapping information to the polished long-read assemblies. Briefly, short reads were mapped using BWA-MEM v0.7.17-r1198 (Li, 2013) to the Medaka-corrected assemblies, and the resulting bam files were used in a single iteration of Pilon correction (using –fix all). (2) Flye assembly: Reads were first assembled into unitigs using the –nano-raw and –meta parameters in Flye v2.5 (Kolmogorov et al., 2019) , with an estimated assembly size of 15M. Medaka and Pilon were then used to correct Flye assemblies as previously described. (3) Hybrid SPAdes assembly: Hybrid assemblies were generated with hybridSPAdes v3.12.0 (Antipov et al., 2016), using the –meta and –nanopore parameters. For comparison and evaluation of error in long-read assemblies, short-read only assemblies were assembled with metaSPAdes v3.12.0 (Nurk et al., 2017) with default parameters. The error-correction performance (number of mismatches and indels) of each long-read assembly/error-correction strategy (and intermediary stages) was assessed with Quast v4.5 (Gurevich et al., 2013) using the -meta option with default parameters against the matching short-read only assembly of the same WEC sample. Genome completeness was computed using CheckV v0.3.0 (Nayfach et al., 2020). Per contig median microdiversity (π) was computed from single nucleotide polymorphism (SNP) frequencies as described previously (Warwick-Dugdale et al., 2019). Briefly, short-reads were first mapped to viral contigs using BWA-MEM with mapping thresholds (70% read coverage at 95% identity) previously established (Gregory et al., 2019) to define viral populations. Of these mapped genomes, only those that had >70% of their length covered by reads, and had >10x coverage were further selected. For each viral population, SNPs were identified using mpileup within samtools (Li et al., 2009) and BCFtools (Danecek & McCarthy, 2017), and low quality variant call (PHRED < 20) were removed. SNPs were identified as ‘true’ if an alternate variant nucleotide was present on at least four mapped reads, and comprised >1% of the base pair coverage for each variant nucleotide at that position.

Generating ‘enhanced’ viromes with short- and long-read assemblies

To produce ‘enhanced’ virome datasets, each assembly type (i.e., metaSPAdes, hybridSPAdes, long-read ‘OLC’) were independently searched for virus contigs (contigs ≥ 2.5 kbp) using VirSorter v1.0.5 (Roux et al., 2015) in –virome search mode. In each assembly type, only viral contigs ≥ 5 kbp belonging to VirSorter categories 1, 2, 4, and 5 were kept. Filtered virus contigs from each assembly were pooled and subsequently dereplicated into viral populations with ClusterGenomes (Roux, 2015), using 70% coverage and 95% nucleotide identity clustering thresholds (sensu (Brum et al., 2015)).

Results and Discussion

Experimental overview

Two sets of experiments were performed (Fig. 1) to evaluate and optimize our long-read laboratory protocol in terms of increased read length, reduced DNA input requirements, and minimized error in assembled genomes. Experiment 1 sought to maximize read length and minimize chimera formation from sheared, amplified DNA extracted from a mock community of cultured phages. This community contained a representative member of the main families of tailed phages (Myoviridae, Siphoviridae, and Podoviridae) and ranged in genome sizes (38.2 kbp–129.4 kbp) and GC content (35.7%–44.7%) (see Methods, Fig. 1A, and Table S1). We evaluated four DNA polymerases (Experiment 1A) using input material sheared to 10kbp; optimized DNA shearing size to generate long-read viromes from a natural seawater sample (Experiment 1B), and evaluated the influence of the number of PCR cycles of chimera formation and read length (Experiment 1C). In Experiment 2 we optimized informatic approaches to maximize viral genome recovery and reduce error using the long-read virome datasets from Experiment 1B in combination with short-reads. We evaluated two assembly/error-correction strategies (Experiment 2A) and benchmarked these against assemblies from short-read approaches, which were assumed to have negligible sequencing error (Experiment 2B).

Figure 1 Overview of wet lab optimization experiments and informatic benchmarking.

(A) Laboratory optimization (‘Experiment 1’) in which a mock community of three phages was used to conduct three experiments aimed at producing longer reads from less input DNA. (B) Informatics benchmarking (‘Experiment 2’) in which a seawater virome was sequenced with short-reads (Illumina) and long-read sequencing (Oxford Nanopore). Three distinct long-read libraries were generated, error-corrected and assembled, and were compared to short-read assemblies to assess accuracy and assembly performance.

Experiment 1A: Amplification with TaKaRA LA yields highest median read lengths

Among the four polymerases tested, amplification with TaKaRa LA Taq yielded the highest median read length (6,965 bp; 6,957–6,973 bp 95% CI). This was a significant increase (p < 0.0001) of 2,866 bp compared to the performance of NEBNext used in the VirION workflow (Warwick-Dugdale et al., 2019), but retained a low proportion of chimeric reads (0.08–0.19%). Therefore, we proceeded with TaKaRa LA Taq for all subsequent tests.

Experiment 1B: Longest read lengths are achieved at low DNA concentrations sheared to 15 kbp

We next sought to evaluate if increased fragment length of 15 kbp provided a concomitant increase in final read length compared to 10 kbp, and whether this was also compatible with low-input DNA concentrations typical of some environmental viromes. Previous work using VirION had shown a discrepancy between expected fragment length (∼8 kbp) and final median read lengths (∼4.1 kbp), suggesting that longer fragments do not necessarily yield significantly longer reads. Final read length may be constrained by the amplicon size produced by the DNA polymerase or preferential amplification of smaller fragments (Shagin et al., 1999; Warwick-Dugdale et al., 2019). Here, shearing 80 ng of input DNA to 15 kbp reduced final median read lengths (6,240 bp; 6,238–6,242 bp 95% CI) in comparison to the 10 kbp shearing treatment (median: 6,965 bp; see Experiment 1A) (Fig. 2B). However, the longest median read lengths were observed when 1 ng of input DNA was sheared to 15 kbp (median: 7,223 bp; 6,899–7,540 bp 95% CI), an increase of 216 bp and 983 bp over shearing 80 ng DNA to 10 kbp and 15 kbp, respectively. There was significant variance in read length between replicates (Fig. 2B), suggesting that: (1) final read lengths may be strongly influenced by variance in downstream library preparation flowcell properties; (2) Interactions between input DNA and centrifugal force selected for shearing may be complex. Both DNA inputs <100 ng and a shear size of 15kbp are outside the official parameters of Covaris g-tube specifications. Therefore, additional optimizations may be required to reduce variance between samples.

Figure 2 Laboratory optimization yield longer reads from less DNA.

(A) Boxplots showing the median and quartiles of the read length distribution between four DNA polymerases. (B) Boxplots showing the median and quartiles of the read length distribution between DNA shearing size treatments and one low-input DNA (1ng) variant of the 15kbp shearing treatment. (C) Boxplots showing the median and quartiles of the read length distribution of three long-read library types, either unamplified or amplified (VirION and VirION2). (D) Boxplots showing the median and quartiles of the read length distribution between four thermocycling treatments (here, number of cycles). Asterisks represent a significant difference (p < 0.0001) between pairs of replicate treatments where applicable.

To assess our revised protocol on real world samples, three long-read libraries (from 14.6 ng of input DNA) from a virus-enriched marine sample were generated and sequenced (Fig. 2C). These libraries were (1) a ‘no amplification’ library (‘unamplified’), (2) our previous VirION protocol, and (3) this study (‘VirION2’). The ‘unamplified’ library had the highest median length (5,601 bp; 5,560 –5,645 bp 95% CI). Between the amplified datasets, the read size distribution from the VirION2 library was significantly higher than VirION. However, in both libraries read sizes were generally lower than those observed in the mock community (at identical shearing size), a phenomenon also observed in our previous study (Warwick-Dugdale et al., 2019). This is likely due to shearing and/or degradation during the FeCl3 precipitation, giving smaller fragments for amplification in the natural sample. Further optimization into maximizing DNA integrity from viral metagenomes could be beneficial to improving long-reads.

Experiment 1C: VirION2 libraries can be prepared with very low input DNA and increased PCR cycles without increasing chimeric reads

Samples with low DNA concentrations require increased numbers of PCR cycles to meet the input requirements (1 µg) of Nanopore sequencing. To maximize VirION2’ s applicability to various sample types, including those with very low input DNA concentrations (∼1 ng), we next tested how amplification cycling (15, 18, 20, and 22 cycles with TaKaRa LA Taq) impacted read lengths and chimeras (Fig. 2D). Starting with the mock community DNA at 1ng concentration, the 20-cycle treatment generated significantly longer reads compared to all other treatments (Wilcoxon rank test, p < 2.22 × 10−16; median = 8,125 bp). Across all treatments and replicates in this experiment, the number of chimeric reads remained near constant (range: 0.01 –0.04%) and therefore, at least within the range tested here, is not an obstacle to increasing cycle number to generate sufficient DNA for sequencing.

Experiment 1 conclusions

Together, the optimization of experiment 1 improve the VirION method in two critical ways. First, the median read length was significantly increased (p < 0.0001) by 76% over the original VirION approach. Second, we were able to generate long-read data from 1ng libraries with a negligible number of chimeric reads, which removed the one microgram requirement for standard MinION libraries, and reduced VirION’s DNA input requirement 100-fold (Warwick-Dugdale et al., 2019). These laboratory protocol improvements should permit a broader diversity of samples to be investigated with long-read sequencing technology. Since Nanopore sequencing is constantly evolving, we have posted the VirION2 method at protocol.io to ensure continued protocol development as the research community identifies new opportunities.

Experiment 2 –Informatics benchmarking

In a second experiment (Fig. 1B), we assessed two informatic approaches for combining long- and short-read data to maximize recovery and accuracy of viral genomes from natural viral communities. A single, virus-enriched seawater sample (‘WEC’; see Methods and Experiment 1B) was used, along with the corresponding Illumina short-reads. Adding short reads served two purposes: (1) as a gold standard for long-read sequencing accuracy estimates and (2) for use in hybrid assemblies and error correction of long-read assemblies.

Experiment 2A: Performance of long-read assemblers and error-correction tools

We first tested an ‘overlap-layout consensus’ (‘OLC strategy’) assembly approach (performed by Miniasm), followed by the successive application of three error-correction tools: Racon, Medaka, and Pilon (the latter uses short-read correction). Of note, assembly of long-read data from this study with Canu (Koren et al., 2017), as used in VirION, (Warwick-Dugdale et al., 2019) failed to finish running within 48 h wall time on an high-performance computing node with 48 CPU cores and 1Tb RAM (data not shown), likely due to the high volume of data generated in contemporary MinION runs (Amarasinghe et al., 2020). In the ‘OLC strategy’, Racon was used to correct raw contigs by generating genomic consensus through multiple iterations of long-read self-mappings, however there is no set standard for how many rounds of Racon to use. Therefore, we first assessed the impact of Racon iterations on error correction by testing 1, 2, 3, 5 and 10 rounds of Racon polishing. Two rounds of polishing were sufficient to produce a sharp decrease (2–3-fold drop) into the number of mismatches (per 100 kbp of sequence), while a single round resulted in a 5 - 8.5-fold reduction in insertion/deletion (‘indels’) events (Fig. S2A). Further rounds of polishing made no significant difference to the number of mismatches and indels removed. Concomitantly, a single round of polishing significantly improved median predicted protein lengths from 71 a.a to 107 a.a but additional rounds resulted in no significant improvements (Fig. S2B). Next, we estimated the effectiveness of error correction between three types of long-read libraries prepared from the same WEC sample (i.e., ‘unamplified’, ‘VirION and ‘VirION2’), compared to short-read only assemblies. Applying Medaka and Pilon provided a moderate reduction (range: 2–13%) of mismatches and a major reduction in indels (range: 70–73%; Fig. S3A) across all three library types. Short-read error correction of long-read assemblies with Pilon had the greatest impact in partially restoring the expected median protein size of 142 a.a predicted from short-read only data (dotted line, Fig. S3B). This was most effective in the VirION2 library, reaching 131 a.a median size (130–132 a.a 95% CI), compared to 113 a.a (112–115 a.a 95% CI) in VirION and 112 a.a (111–113 a.a 95% CI) in the ‘unamplified’ dataset. Therefore OLC assembly of VirION2 reads, coupled with two rounds of Racon polishing, followed by one round of Medaka and additional short-read polishing with Pilon is our recommended approach, and yielded the most improved assembly correction between the three long-read libraries, as well as reaching median protein size closest to the short-read assembly.

We next tested the effectiveness of assembly by repeat graphs using Flye (see Methods) to process our virome. Recent reports (De Maio et al., 2019; Moss, Maghini & Bhatt, 2020) suggest good performance in terms of assembly accuracy and scale in prokaryotic datasets, however its performance with viromes remains untested. Assembly of WEC long-read datasets with Flye was followed by the same error-correction tools as described for ‘OLC strategy’, except for Racon (Racon is designed for Miniasm only, and Flye has an internal error-correction module). Despite repeated attempts and large computing allowances (48 CPU cores/1Tb RAM), we could not generate a Flye assembly from the ‘unamplified’ library. This could be due either to sequence complexity in the sample, which can complicate estimating the required genome size parameter of the program, and/or insufficient resources to run (Wick & Holt, 2020). However, both VirION and VirION2 datasets were successfully assembled by Flye without large resources, therefore we focused our analysis on these. Similar to the OLC strategy, indel reduction was more successful (range: 74–75% removed) compared to mismatches. Surprisingly, mismatches in Flye-assembled contigs marginally increased with polishing by Medaka –a feature that was not entirely corrected with subsequent application of short-read polishing with Pilon, resulting in an overall 1.25% removal in VirION2 only, but an increase of 4.53% in VirION) (Fig. S3C and Table S2). Corrected Flye assemblies had shorter median protein sizes compared to the OLC strategy results in both VirION and VirION2 datasets (by 5 and 21 amino acids, respectively, Fig. S3D), perhaps due to greater indel reduction in the OLC strategy. Therefore, in terms of correction performance, the OLC strategy is preferred over the Flye strategy, at least for assembly of long-read viral metagenomes with current software versions.

Experiment 2B: VirION2 coupled with OLC assembly provides greatest gains in assembly and predicted gene lengths

Using the results generated in experiment 2A, we compared all the corrected long-read assembly types to hybrid assemblies (i.e., short read assemblies scaffolded by long-reads) and short-read only assemblies (Fig. 3). Across all assembly types (hybrid, OLC and Flye) and libraries (unamplified, VirION and VirION2), sequence accuracy of polished long-read assemblies ranged from 99.61% to 99.98% (Table S2). The VirION2 OLC strategy offered an increased accuracy of ∼0.05% over ‘unamplified’ and VirION-amplified reads. Although seemingly small, this increased accuracy translated into a much-improved median protein size of 131 a.a (130–132 a.a; 95% CI), compared to 112 a.a (111–113 a.a; 95% CI) in the unamplified library and 113 a.a (112–115 a.a; 95% CI) for the VirION library. Regardless of library method, Flye assemblies consistently yielded the lowest median protein sizes. In terms of contigs lengths, N50 metrics were markedly improved (∼4-fold) in all assembly strategies in comparison to short-read only assemblies (Table S3). Between VirION and VirION2 specifically, the N50 from VirION2 contigs was superior in both OLC (31,496 bp versus 16,840 bp) and Flye strategies (31,130 bp versus 21,231 bp). The same was true for maximum contig sizes reached: 194,588 bp versus 115,168 bp (OLC), and 570,045 bp versus 140,205 bp (Flye). Overall, between library preparation (unamplified, VirION and VirION2) and long-read assembly/correction strategies (OLC and Flye), combining VirION2 with the OLC strategy performed best (99.71% accuracy, Table S2) and is the preferred option for accurate long-read virome datasets, with the critical advantage of lowering the input DNA requirement by 100-fold.

Figure 3 Error-correction profiles between library methods and assembly strategies using the WEC sample.

(A) On the x-axis, mismatches and insertion/deletion (‘indels’) events according to assembly strategy (full OLC, full Flye, and Hybrid) are grouped separately, and divided into three facets, one for each long-read library method (unamplified, VirION, VirION2). The number of errors (y-axis) is scaled to the binary logarithm (log2) for scale fitting purposes. (B) Boxplot depicting the protein size distribution (in amino acids, denoted as ‘a.a’; y-axis) derived from each library method (x-axis), each of which is sub-grouped per assembly strategy. In both A and B panels, there were no results from Flye assemblies for the raw datasets, as these could not be produced.

Figure 4 Comparison of virus genome properties between short-read and ‘long-read-enhanced’ viromes.

(A) Workflow to produce ‘enhanced viromes’, in which Spades, hybrid and long-read (OLC) viruses are combined to maximize the recovery of virus signals. (B) Cumulative Distribution Function (CDF) plot depicting the frequency (y-axis) of virus genomes according to genome length (measured in kilo basepairs (kbp), x-axis) between three assembly strategies. (C) Cumulative Distribution Function (CDF) plot depicting the frequency (y-axis) of virus genomes according to genome ‘completeness’ (measured in %, x-axis) between three assembly strategies. (D) Cumulative Distribution Function (CDF) plot depicting the frequency (y-axis) of virus genomes according to genome microdiversity per genome (measured as π , x-axis) between three assembly strategies.

Figure 5 Additional insights are gained through a VirION2-enhanced assembly strategy.

(A) Rank abundance curve depicting the seawater virus community, colored according to whether a virus population (individual bars) was detected uniquely (turquoise) or in multiple (pastel red) assembly types. The top 10% most abundant viral populations are highlighted between dashed lines, where they are divided per assembly origin in the pie chart. (B) Boxplots depicting the level of microdiversity between shared and unique viral populations within each constituent assembly present in the ‘enhanced’ dataset.

VirION2-enhanced viromes recovers longer and more microdiverse viral populations

Next, we evaluated how well short- and long-read assemblies recovered viral populations from a natural viral community, in terms of genome size, genome completeness and microdiversity (i.e., intra-population genomic variation). Specifically, we compared a short-read-only virome against ‘enhanced’ viromes that maximize the benefits of both short- and long-read technologies (derived from combining multiple assemblies from both read types; see Methods and Fig. 4A). Genome size, genome completeness, and microdiversity were all significantly improved in VirION2-enhanced viromes, compared to short-read and VirION-enhanced viromes (Figs. 4B–4D and Table S4). Notably, recovery of >100 kbp genomes in the VirION2-enhanced virome was increased by 13- and 2-fold (Fig. 4B), relative to short-reads only and VirION-enhanced, respectively. Accurate and contiguous assembly of large phage genomes could increase the detection of ‘jumbo’ phages throughout ecosystems, as there is currently limited data on the distribution and genome diversity for large phages (Yuan & Gao, 2017; Al-Shayeb et al., 2020). Microdiversity (measured as π, Fig. 4D) increased by 2.4–2.5-fold in both VirION- and VirION2- enhanced viromes, respectively, compared to the short-read assemblies. A similar increase was reported in the original VirION method from WEC seawater samples (3-fold; Warwick-Dugdale et al., 2019). However, there was no significant difference between VirION and VirION2 datasets (Mann Whitney U-test, p-value = 0.536). Although it is tempting to speculate that residual sequencing error in long-read datasets could artificially increase SNP frequency and therefore higher microdiversity (π) values, we found that its impact is negligible due to the error-correction pipeline applied, along with the error distribution patterns (mostly indels) of Nanopore reads (for more details, see Analysis S1). Alternatively, the higher microdiversity in ‘enhanced’ datasets compared to short-read data could be due to novel viral populations uniquely captured with long-reads (discussed below). In addition, it is worth noting that during the multiple rounds of polishing of long-read assemblies, genuine biological diversity may be removed from consensus sequences, leading to an underestimation of microdiversity.

Long-read viromics improves capture of abundant viral populations compared to short-read only methods

In addition to better genomic metrics, we assessed how the ‘VirION2-enhanced’ assembly from the WEC seawater sample could alter our biological interpretations of complex viral communities. We first evaluated if more viruses could be detected by adding long-reads, and by counting the number of shared and unique viruses in each constituent assembly that make the VirION2-enhanced virome (i.e., long-read only, short-read only, and hybrid). Among the total number of viral populations (n = 5, 161), 68% were represented in all assembly types. Within the pool of viral populations observed only within a single assembly method, populations from the hybrid assembly were the majority (23%), followed by the OLC assembly (7%), and short read assembly (2%). We surmise that adding long reads permitted a more complete view of a virome, and viral genomes that would otherwise have been missed by short-read assembly alone.

Next, we looked at patterns in abundance of viral populations as a function of assembly method within the VirION2-enhanced virome (Fig. 5A). Among all viruses, the majority of these unique viruses were low abundance populations, and were short fragments from the hybrid assembly. However, within the top 10% most abundant viruses (n = 516), 164 were unique to a particular assembly method and 70% of these were derived from the long-read only assembly. Taken together, the addition of long-reads uniquely enabled the recovery of 22% of the most abundant viruses—that would have been missed in short-read only assemblies. We queried whether microdiversity could be a contributing factor as to why these unique viruses would be missed. Indeed, when we looked at the fraction of ‘unique’ viral populations within each assembly type in the ‘enhanced’ virome, unique viral populations had significantly higher microdiversity compared to the shared pool of viruses in each assembly, most predominantly observed in the ‘OLC’ assembly (Fig. 5B). Thus, our data suggest that high intrapopulation nucleotide variation in a subset of virus genomes can lead to these genomes being fragmented (Roux et al., 2017) and overlooked in a short-read only viral metagenome experiment.

Current limitations and future directions

Despite the advances of VirION2, several limitations remain. First, an assembly step (due to the required shearing of input DNA), as well as corresponding short-reads, remain a necessity to ensure the highest level of error-correction, and to enable microdiversity estimates. Ongoing developments that increase recovery of full-length genomes as single long-reads and optionally use short-reads for correction (Beaulaurier et al., 2020) will certainly help. However, this workflow cannot be used for low-input samples, as it requires micrograms of high molecular weight DNA to yield full genomes as single reads. Moreover, even if sufficient DNA is obtained, our tests using ‘unamplified’ datasets indicated that compared to the VirION2 approach, viral populations from the unamplified library had shorter median proteins sizes (112 a.a versus 131 a.a, respectively). Second, the revised long-read library protocol was designed and evaluated for the recovery of dsDNA virus genomes. Future refinements could include modifying the current protocol for the inclusion of ssDNA and RNA genomes (through direct RNA or cDNA libraries). Third, due to the amplification step within the VirION2 laboratory protocol, we cannot leverage base modification capabilities from Nanopore sequencing, as these can only be detected if intact, non-amplified high molecular weight DNA is used as template for sequencing. Lastly, we have not evaluated VirION2 on the latest R10 flowcells, which promises further sequencing accuracy, especially within homopolymeric regions. Adapting VirION2 to R10 chemistry (R9.4 flowcells were used in this study) to generate long-read viromes, as well as sequencing depth available from the PromethION (the higher throughput version of the MinION sequencing platform) remain to be tested, but we do not anticipate major protocol (either wet-lab or informatics) modifications will be required. Lastly, additional laboratory optimizations parameters could be tested, including whether further increases in DNA shearing sizes (e.g., 20kbp or higher) may further improve long-read sizes, albeit with likely diminishing returns.

Conclusions

Though short-read sequencing has become the gold standard in viral metagenomics, it is increasingly clear that this approach does not capture the full extent of virus macrodiversity (i.e., species richness) and microdiversity (i.e., intra-population genomic diversity). The latter in particular, can cause genome fragmentation and mask genes under active selection pressures (e.g., genomic islands) and important indicators of virus-host dynamics. Long-read sequencing, combined with short-reads, can further increase estimates of viral diversity and capture ecologically important taxa. Owing to the substantial decrease in DNA requirements for VirION2, long-read viral metagenomics may now be applied to a broader range of samples, thus constituting an invaluable addition to our current viral ecogenomics toolkit for the better exploration of viromes and their impacts in nature.

Supplemental Information

Supplemental Information 1 Visualizing sub-sampling impacts on long-read size datasets

Frequency histograms depicting the read size distribution between the three long-read libraries: ‘raw’ (i.e., unamplified input DNA), VirION, and VirION2. Each library type is represented in its own facet, with both full (pastel red) and sub-sampled (turquoise) size distributions overlapping. Dotted lines and their associated colors denote the median value of each distribution.

Click here for additional data file.

Supplemental Information 2 Impact of error-correction with the Racon tool using VirION2 data from a seawater sample (‘WEC’)

(A) Bar chart depicting the number of mismatches and insertion/deletions (indels), grouped by number of Racon rounds (on the x-axis) across increasing iterations of Racon polishing of long-read assemblies. From light to dark blue, increasing color saturation corresponds to the increasing number of Racon polishing rounds (also applicable to panel B). (B) Boxplot depicting the impact of consecutive rounds of Racon (x-axis) on predicted protein sizes (y-axis, measured in amino acids denoted as ‘aa’). The horizontal dotted line represents the median protein size of the corresponding short-read assembly (142 amino acids).

Click here for additional data file.

Supplemental Information 3 Error-correction levels at each correction stage within OLC and Flye strategies compared between long-read library methods

(A) Bar chart depicting the number of mismatches and insertion/deletions (indels) from the OLC strategy, grouped according to increasing correction (on the x-axis), separated by library method (raw, VirION, VirION2). (B) Boxplots depicting protein size distribution at each level of correction in the OLC strategy. The horizontal dotted line represents the median protein size of the corresponding short-read assembly (142 amino acids). (C) Bar chart depicting the number of mismatches and insertion/deletions (indels) from the ‘Flye’ strategy, grouped according to increasing correction (on the x-axis), separated by library method (VirION and VirION2). (D) Boxplots depicting protein size distribution at each level of correction in the ‘Flye’ strategy. The horizontal dotted line represents the median protein size of the corresponding short-read assembly (142 amino acids).

Click here for additional data file.

Supplemental Information 4 Per-genome microdiversity distributions within the VirION2-enhanced viromes constituent assemblies

Boxplots displaying the per-genome microdiversity distribution across each virome assembly type (i.e.”,Short-reads only”, ”VirION-enhanced”, ”VirION2-enhanced”). Except for the ”Short-read only assembly”, data is further divided according to the origin of each constituent assembly (i.e., “Hybrid”, “OLC”, “Spades”).

Click here for additional data file.

Supplemental Information 5 Evaluation of the impact of sequencing error on microdiversity estimates

Click here for additional data file.

Supplemental Information 6 Summary characteristics of the virus isolates used in the mock community for Experiment 1

Click here for additional data file.

Supplemental Information 7 Error-correction performance benchmarks between assembly methods and sequencing libraries

Click here for additional data file.

Supplemental Information 8 Contigs and protein statistics between assembly strategies above 2.5 kbp

Click here for additional data file.

Supplemental Information 9 Genome-based metrics between short-reads-only and virION-enhanced datasets (¿5 kbp genomes)

Click here for additional data file.

Supplemental Information 10 Tracking SNP and microdiversity detection in mock community long-read assemblies at different stages of error correction

Click here for additional data file.

The authors thank the crew of the Plymouth Marine Laboratory vessel ‘Quest’ for collection of seawater samples. Research was supported by high performance computing resources provided by the University of Exeter and the Ohio Supercomputer Center.

Additional Information and Declarations

Competing Interests

Author Contributions

DNA Deposition

Data Availability

The authors declare there are no competing interests.

Olivier Zablocki conceived and designed the experiments, analyzed the data, prepared figures and/or tables, authored or reviewed drafts of the paper, and approved the final draft.

Michelle Michelsen and Joanna Warwick-Dugdale performed the experiments, authored or reviewed drafts of the paper, and approved the final draft.

Marie Burris and Natalie Solonenko conceived and designed the experiments, performed the experiments, authored or reviewed drafts of the paper, and approved the final draft.

Romik Ghosh analyzed the data, prepared figures and/or tables, authored or reviewed drafts of the paper, and approved the final draft.

Jennifer Pett-Ridge and Matthew B. Sullivan conceived and designed the experiments, authored or reviewed drafts of the paper, and approved the final draft.

Ben Temperton conceived and designed the experiments, performed the experiments, analyzed the data, prepared figures and/or tables, authored or reviewed drafts of the paper, and approved the final draft.

The following information was supplied regarding the deposition of DNA sequences:

The mock community experiments and the Western English Channel virome sequencing datasets are available at the European Nucleotide Archive: PRJEB41030.

The following information was supplied regarding data availability:

The WEC assemblies are available in the Supplemental Files.

Protocol is available on protocols.io:

Marie Burris, Ben Temperton, Natalie Solonenko 2020. VirION 2. protocols.io https://dx.doi.org/10.17504/protocols.io.6q9hdz6.

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
