# Peer review of "VirION2: a short- and long-read sequencing and informatics workflow to study the genomic diversity of viruses in nature"

_PeerJ, doi:10.7717/peerj.11088_

## Round 0.1 · original submission · Major Revisions

Both reviewers (and I) recognize the value of this work, and believe that it will make a significant contribution to the discipline. However, based on the reviews, it is apparent that some clarifications are required in order for this manuscript to meet its potential.

Reviewer 1 ·

Basic reporting

Article is well written. Language is clear and coherent. The study relies fairly heavily on precedent set by the initial workflow "virION." While this is understandable (as a methods-based comparative study), it is a bit difficult to read as a standalone. I think it may benefit from a brief description of the original virION workflow with relevant references or figure comparing workflows. Without this context, it is a bit difficult to justify why improvement on the virION pipeline is needed and determine if this is an appropriate comparative study without reading the other paper.

Will sequencing data or assemblies eventually be deposited somewhere publicly, and if so, is there a place in the manuscript designated for an accession number or other affiliated access number? Will PRJEB41030 be explicitly reported? As a living document at protocols.io, is there version control ensuring a static copy of this protocol through its continued evolution?

Very minor:
- L65 unclear which definition of “strain” authors utilize - antigenically different? Most dominant variant? It’s a little distracting and could be either changed (e.g. genotype, variant, lineage, ?) or clarified (I believe both references here utilize bacterial strain microdiversity definitions which are a little confusing)
- L41 in the abstract, it is unclear if “microdiversity” reporting is a mean or median; L78 perhaps specify “complete viral genomes?”; L82 “reduce input DNA requirements”?; L97 manufacturer kit reference would be helpful here ; L102 and 115 capitalization
- L124 - ONT - oxford nanopore is not identified as ONT before this, to my knowledge? Also perhaps specifically define that CS is a positive control step? (is it?). on L176, it may also be helpful to redefine ONT to reiterate short v. long-read
- L263 “may be strongly,” L269 “marine ecosystem?” L271 median length?, L449 intra-population diversity? Intra-genomic diversity? L443 throughput?

Experimental design

This is a methods-based study investigating techniques to merge long-read and short-read sequencing platforms to better capture viral population microdiversity and viruses with large genomes. The study examines both wetlab and computational approaches to optimize this integration, which the authors named virION2, a pipeline that improves on a previously benchmarked workflow, virION.
This study is original, though predicated on the first pipeline, and (to my knowledge) within the scope of the journal. Methods are described in detail and workflow included on protocols.io. Additional clarification of short-read libraries, sequencing platforms, virION “original” (to clarify which improvements have been made), etc would be very helpful, as noted below to determine if the experimental design is appropriate. The study provides important insight into the effectiveness of certain tools.

Validity of the findings

From my understanding, this is a methods & benchmarking-based study. The workflow appears tractable and well executed. However, I’m confused if it assists in solving the field-wide “problem” that it originally aims to address in the introduction: current platforms miss microdiversity and large viral genomes, and this platform hopes to assist in this. It improves on the original pipeline virION by reducing DNA input and increasing read length, among many other aspects. However, because it still needs large quantities of DNA input to assess microdiversity and resolve large genomes, it doesn’t appear to solve the argument I believe the authors pose? Some additional clarification in the “current limitations” might help me understand a bit better.

That being said, the benchmarking presented here is inherently valuable to both wetlab and computational virologists. It serves as a great reference to evaluate the utility of certain polymerases, kits, and programs. Thank you for this!

Specific comments:
- L192 & paragraph: is it possible to justify why libraries were normalized by number of reads, rather than another parameter (e.g. #bases), when read length is inherently going to be highly variable based on platform/workflow? Is it possible that this could significantly alter reports of genome completeness, length, SNV calling, etc?
- I may have missed this, but I cannot find anywhere in the main manuscript how mock short-read libraries were prepared or sequenced. I assume they were prepared using a nextera kit (#?) as “nextera” is included in supplementary figures. I also assume that they were sequenced 2x125bp PE on a HiSeq 2500, because this is how WEC samples were sequenced? However, because this is a methods-based paper, it is difficult to know what SNVs are introduced in this step without reporting.
- L244 Can you clarify the prioritization of length over high-fidelity polymerases, when assessing microdiversity appears to be a central goal?
- Though input viral DNA volumes or concentrations were sometimes reported, were they ever QC’d? If so, what device was utilized?
- Authors indicate that increased SNP frequency in longer read assemblies such as virION is not due to sequencing platform (L392). In error-correction, the multitude of amplification steps - which likely introduce SNPs - are also likely controlled for, correct? Does the potential for uneven recruitment of “unamplified” libraries (is this a non-PCR nextera short-read library?) alter error correction in these longer-read “enhanced” viromes, and if so, does this impact assessments of microdiversity? Would it be possible to clarify these in the manuscript?
- The conclusions must be read very carefully when juxtaposed with the limitations paragraph directly above and authors could clarify this for the benefit of their readers. It appears that with low DNA input requirements, virION2 will not assess microdiversity or capture viruses with large genomes (large DNA input is still required for error correction), and this is a subtly that could be easily missed.

Additional comments

These bioinformatic comparisons are highly valuable and exciting to read. These types of studies tend to be thankless and complicated, but the authors do an excellent job of accounting for multiple parameters. However, insufficient reporting (and a convoluted presentation of conclusions) makes it somewhat difficult to determine if virION2 is appropriately benchmarked and ultimately applicable to the scientific community.

Reviewer 2 ·

Basic reporting

The work is well written with clear rational for the work. The raw data is provided via databases


The use of the virION2 method, clearly improves the recovery of vial genomes by combining long and short read sequencing. This, improvement has been observed before when combing long and short read methodologies for viral sequencing. Although using different method for the production of long reads, an improvement on short read assembly alone has been observed previously. These methods really should really be mentioned in the introduction (https://doi.org/10.1101/2020.10.08.329714. PMID: 32761733 ) to give a complete picture of the current approaches that have used long reads in viromics. Given they had similar conclusions

Experimental design

The research question is well defined. The experiments are comprehensively carried out, with a large number of variables tested for the optimization of the method. It meets a clear need to provide reproducible methods for the production of long read viral metagenomes from a low input DNA. It will be of great use to the viral community

Methods are generally very well described. It would be helpful to include the method for micro diversity in this paper, given it is an essential parameter. Rather than referring to a previous work where it is in the supplementary materials.

Validity of the findings

The conclusions are well supported by the data presented

Additional comments

The described work builds on previous work by some of the authors on the use of minION sequencing and its use for viral metagenomics. The work comprehensively and robustly optimises the use of the “VirION” method for long read viral metagenomics. Demonstrating that a combination of both long and short read sequencing improves the quality of assembled contigs. Such method optimisation approaches and bench marking are extremely useful tools for the wider community. The paper is very well written; with all data accessible and the methods fully described, (one case could do with further details being provided in this paper). The work was a pleasure to read and highlighted the importance of this improved approach

Minor points
L79 – here and elsewhere, space between units and number 20 L
L220 – given the importance of micro diversity to this study, it would be helpful if the method was provided in the work too. Rather than readers having to refer to the supplementary file of previous publications.

---

## Round 0.2 · accepted · Accept

I enjoyed reading this manuscript, and agree with the two reviewers that this is a valuable contribution to the discipline.